# Vaccine-Induced Immune Thrombotic Thrombocytopenia following BNT162b2 mRNA COVID-19 Booster: A Case Report

**DOI:** 10.3390/vaccines11061115

**Published:** 2023-06-19

**Authors:** Tzu-Chien Lin, Pei-An Fu, Ya-Ting Hsu, Tsai-Yun Chen

**Affiliations:** 1Department of Internal Medicine, National Cheng Kung University Hospital, College of Medicine, National Cheng Kung University, Tainan 704, Taiwan; chiang0314@gmail.com (T.-C.L.);; 2Division of Hematology, Department of Internal Medicine, National Cheng Kung University Hospital, College of Medicine, National Cheng Kung University, Tainan 704, Taiwan

**Keywords:** vaccine-induced immune thrombotic thrombocytopenia, mRNA vaccine, BNT162b2, COVID-19 vaccine booster

## Abstract

Vaccine-induced immune thrombotic thrombocytopenia (VITT) is a life-threatening complication caused by platelet activation via platelet factor 4 (PF4) antibodies. We report a healthy 28-year-old man who developed hemoptysis, bilateral leg pain, and headaches three weeks after his third dose of the COVID-19 vaccine with the first BNT162b2 (from Pfizer-BioNTech) injection. He had previously had the first and second doses with ChAdOx1 nCov-19 without any discomfort. Serial investigations demonstrated pulmonary embolisms, cerebral sinus, and deep iliac venous thrombosis. Positive PF4 antibody assay (ELISA) confirmed the diagnosis of VITT. He had a prompt response to intravenous immunoglobulins (IVIGs) at a total dose of 2 g/kg and his symptoms are now in remission with anticoagulant. Although the definite mechanism is unknown, the VITT was most likely triggered by his COVID-19 vaccine. We report this case of VITT following BNT162b2, a mRNA-based vaccine, and suggest that VITT could still happen without the adenoviral vector vaccines.

## 1. Introduction

The COVID-19 pandemic, resulting from the infection of the SARS-CoV-2, has had a global impact, prompting extensive vaccination as a vital strategy to mitigate its effects. In order to control the pandemic, the scientific community has made significant progress in mitigating the threat of COVID-19 through the discovery and development of vaccines [1], small molecule agents [2], antibodies [3], natural products [4], and traditional medicines [5]. Clinical trials of COVID-19 vaccines have yielded compelling evidence supporting their overall safety and efficacy [6,7,8,9,10]. However, recent reports have brought attention to a rare but consequential complication associated with vaccination, namely vaccine-induced immune thrombotic thrombocytopenia (VITT). This complication necessitates a thorough understanding of its clinical features, underlying mechanisms, and appropriate management strategies to ensure continued safety.

VITT is characterized by a unique combination of thrombocytopenia (low platelet count) and thrombotic events, which manifests as thrombosis occurring in unusual locations [11,12]. In this condition, the excessive activation of platelets, coupled with dysregulated coagulation processes, creates a hypercoagulable state, predisposing the individual thrombosis in both venous and arterial vessels, as well as secondary hemorrhage. These events typically occur within 5 to 30 days following vaccination and can lead to life-threatening complications [11]. One of the key diagnostic markers for VITT is the presence of antibodies against platelet factor 4 (PF4) in the patient’s blood, along with the detection of PF4 bound to platelets [13]. These PF4 antibodies activate platelets through low-affinity platelet Fcγ receptors, contributing to the development of VITT [14]. Although VITT is considered a rare condition, the reported incidence ranges from 3 to 15 cases per million vaccinations [11,15]. It is important to note that the actual incidence may be higher due to under-recognition and under-reporting. Most cases of VITT have been associated with vaccines utilizing a recombinant adenoviral vector encoding the spike protein antigen [14]. The ChAdOx1 nCoV-19 (AstraZeneca) vaccine, in particular, has been linked to the majority of reported VITT cases [14,16].

In this report, we present a rare case of VITT in a previously healthy young man who developed the condition after receiving the BNT162b2 (Pfizer-BioNTech) vaccine. The patient experienced pulmonary embolism, cerebral sinus thrombosis, and deep iliac venous thrombosis, underscoring the potential severity and multi-organ involvement associated with VITT.

## 2. Case Report

A previously healthy 28-year-old male patient arrived at the emergency department (ED) with sudden bilateral leg pain and several pinhead-sized purpuric papules on his legs. He had experienced occasional chest pain, hemoptysis, abdominal pain, and headaches for three weeks prior to his ED visit. These symptoms had developed about three weeks after receiving his third dose of the BNT162b2 (Pfizer–BioNTech) COVID-19 vaccine in January 2022, and all of them were self-limited. He had received two doses of the ChAdOx1 nCoV-19 vaccine nine and seven months prior without any adverse events. The patient had no risk factors or family history of thromboembolic diseases, and there was no recent history of travel. He did not smoke or use recreational drugs. The patient had not been administered any pharmacological treatment regimen. The physical examination revealed bilateral leg mild edema and right leg numbness. There were no findings of hepatosplenomegaly upon palpation of the abdomen, and the patient did not exhibit any signs of respiratory distress. Upon arrival, he had a fever but stable hemodynamics. The results of his blood tests revealed thrombocytopenia, with a platelet count of 32 × 10^9^/L, elevated D-dimer levels (>7650 ng/mL) (Table 1), and negative results for lupus anticoagulants and antiphospholipid antibodies. He also tested negative for COVID-19 infection. Considering the patient’s clinical presentation, which included the presence of multiple papules on the legs and the occurrence of multi-organ symptoms, it is noteworthy that serial blood tests had yielded no evidence supporting the diagnosis of autoimmune vasculitis. These results suggested that an alternative etiology may be responsible for the observed clinical manifestations. Thus, further comprehensive investigations were conducted. A contrast-enhanced chest computed tomography (CT) scan revealed pulmonary embolism and partial thrombosis in the inferior vena cava. CT venography of the lower extremities revealed deep venous thrombosis (DVT) within the bilateral common iliac veins (Figure 1a,b). Brain magnetic resonance imaging (MRI) showed cerebral sinus thrombosis with concurrent subacute intracerebral hemorrhage (ICH) (Figure 1c,d). A final PF4 antibody assay (ELISA)/heparin enzyme-linked immunosorbent assay confirmed a positive result (80.26 ng/mL; Optical Density value: 0.793) (Biotek Synergy HTX multimode reader, CA, USA; reference value, Optical Density value < 0.4), confirming the diagnosis of VITT.

Taking into account the immunological-driven thrombotic nature of the disease, the initial treatment for the patient involved the administration of methylprednisolone at a dose of 1 mg/kg/day. However, the effect of corticosteroid was limited. Once the confirmation of VITT, the patient received a therapeutic intravenous immunoglobulins (IVIGs) administered at a dose of 1 g/kg over a two-day period. Simultaneously, the patient was initiated on dabigatran, an oral anticoagulant, immediately. Given his post-thrombotic cerebral hemorrhage, repeated head CT scan showed a stable left temporal lobe ICH. The patient experienced a favorable clinical outcome without any new neurological symptoms. Additionally, his platelet count demonstrated a progressive normalization over the course of treatment (Figure 2). He was eventually discharged with a good recovery. Over a month, he demonstrated a complete recovery from thrombocytopenia and the resolution of neurological symptoms. On follow-up, he remained well while on Dabigatran for six months, with a normal hemogram.

## 3. Discussion

We present a case study of a 28-year-old male patient who developed VITT approximately one month after receiving a first booster dose of the COVID-19 BNT162b2 vaccine. The patient’s clinical course was complicated by cerebral venous thrombosis, intracranial hemorrhage, DVT, and pulmonary embolism. To our knowledge, there is limited documentation of VITT associated with such catastrophic multiple thrombosis in relation to the Moderna BNT162b2 vaccine. Consequently, we conducted a confirmatory diagnosis of VITT using a PF4-induced platelet activation assay.

As of 2022, COVID-19 continues to pose a significant global health concern. The most effective strategy in controlling the pandemic is through widespread immunization efforts. Millions of people worldwide have received COVID-19 vaccines [17], and their safety has been well-established. Initially, adenoviral-vectored DNA vaccines were associated with the rare thrombotic condition known as VITT, which shares similarities with heparin-induced thrombocytopenia (HIT). In HIT, the interaction between negatively charged heparin and positively charged PF4 plays a key role. The ChAdOx1 nCoV-19 vaccine, utilizing a viral vector derived from the chimpanzee adenovirus, is known to induce VITT. The clinical and laboratory resemblances to HIT are substantiated by the identification of high-titer antibodies to PF4 in the serum of affected individuals, which elicit platelet activation. Unlike HIT, where antibodies predominantly bind to the PF4-heparin complex, in VITT, anti-PF4 antibodies mainly bind to PF4 alone. In VITT, the ChAdOx1/PF4 complex may induce the production of anti-PF4 autoantibodies. Following intramuscular vaccine administration, trace amounts of ChAdOx1 may enter the bloodstream due to slight capillary damage [18]. This may lead to the formation of a ChAdOx1/PF4 complex, triggering the production of autoantibodies. Another possibility is that PF4 may bind to the negatively charged adenoviral vector, thus becoming immunogenic and causing the production of autoantibodies against PF4. PF4 can also bind to other polyanions such as heparin, DNA, and even bacteria [19]. Although the adenoviral vector is the primary reported mechanism in VITT, other potential mechanisms include molecular mimicry, contamination of the vaccination protein, the presence of EDTA buffers, or immunity to the viral spike protein [20]. In our case, the occurrence of VITT following BNT162b2 vaccination suggests alternative mechanisms other than those associated with adenoviral-vectored vaccines. The patient’s normal laboratory tests and absence of symptoms prior to vaccination raises the possibility of an association between VITT and the BNT162b2 vaccine.

COVID-19 messenger RNA (mRNA) vaccines have undergone comprehensive assessment in the adult population, revealing their safety and efficacy as potent interventions for mitigating the incidence of SARS-CoV-2 infection, attenuating disease severity, and ameliorating persistent post-COVID sequelae [6,8,21,22,23]. Until now, there have only been a limited number of reported instances of VITT attributed to mRNA COVID-19 vaccines, such as mRNA-1273 (Moderna) or BNT162b2. Chen et al. documented a case study detailing VITT characterized by cerebral venous thrombosis and hemorrhage subsequent to the administration of an mRNA-1273 COVID-19 booster 32 days ago [24]. The patient had previously received ChAdOx1 nCoV-19 vaccinations seven months and four months prior. In contrast to our patient, this particular case exhibited refractory thrombocytopenia that was unresponsive to IVIGs therapy, ultimately resulting in the patient’s demise due to irreversible brain injury. Aside from the mRNA booster-induced VITT, Rodríguez et al. reported a case of VITT developed 25 days after his first dose of the BNT162b2 vaccine [25]. The clinical symptoms rapidly resolved after direct oral anticoagulant. In contrast to our case, this patient did not receive IVIGs therapy as part of their treatment. It appears that both the first dose and the booster of the mRNA vaccine have the potential to cause this rare complication.

As outlined by Pavord et al., the conclusive diagnosis of VITT should only be established if all five specific criteria are met. Firstly, symptoms must manifest within 5 to 30 days following the administration of the SARS-CoV-2 vaccine. Secondly, there should be clear evidence of thrombotic events occurring. Thirdly, thrombocytopenia, characterized by a low platelet count, must be observed. Fourthly, elevated levels of D-dimer, a fibrin degradation product, should be detected. Finally, the presence of positive anti-PF4 antibodies, confirmed by enzyme-linked immunosorbent assay (ELISA), further supports the diagnosis of VITT. The presentation of our patient fulfills all the diagnostic criteria for VITT. Cerebral veins are the most common site of thrombosis in VITT, and up to 36% of cases are complicated by secondary intracranial hemorrhage [11]. In addition to cerebral venous thrombosis, the patient in this case also had thrombosis in the deep veins of the legs and pulmonary arteries, which are the second most common sites of thrombosis in VITT [11]. Overall, the clinical presentation of our patient aligns with VITT induced by adenoviral-vector vaccines.

In the year 2021, there were documented instances of uncommon thrombotic events accompanied by thrombocytopenia after COVID-19 vaccination. Notably, all reported cases were associated with vaccines that utilized a recombinant adenoviral vector, encoding the spike protein antigen [14]. As previously mentioned, following the intramuscular administration of the vaccine, a cascade of events may ensue, including microvascular damage, hemorrhage, activation of platelets, and the release of PF4, which is one of the proposed mechanisms of VITT. Benefitting from knowledge through previously reported VITT cases and the evolving treatment references [26,27], our patient promptly received high-dose IVIGs and therapeutic oral anticoagulation. It is important to note that prophylactic platelet transfusions are not recommended in this particular condition. After treatment initiation, the patient exhibited gradual improvement in multiple thrombotic events, secondary intracerebral hemorrhage, and associated symptoms. Although the occurrence of VITT in our patient was not attributed to an adenoviral vector vaccine, the proposed treatment strategies still proved to be effective in this situation.

With regard to the mRNA COVID-19 booster vaccination, a recent study demonstrated the safety of mRNA boosters following ChAdOx1 nCoV-19 vaccination in cases of typical VITT presentation [19]. Most platelet-activating anti-PF4 antibodies in VITT are transient, and the median time to negative functional test results was found to be 15.5 weeks [28]. Although seroconversion of negative anti-PF4/heparin IgG enzyme immunoassay (EIA) was observed in 21.5% of patients after a median follow-up of 25 weeks, none of the VITT patients developed new thromboses or a rise in anti-PF4/heparin IgG EIA optical density after receiving the mRNA COVID-19 vaccine booster [28]. Therefore, mRNA vaccine boosters are generally considered safe for VITT patients. In a previous study about VITT, platelet-activating anti-PF4 IgG antibodies decline over time in the majority of patients, indicating that the condition tends to resolve on its own [14,29]. Nevertheless, there are some patients who maintain elevated levels of these antibodies, invoking questions as to whether VITT could become a persistent condition called long VITT. This scenario has implications for treatment regarding the duration of anticoagulation and the potential need for immunosuppressive agents. There have been reported cases of long-duration VITT where patients continue to exhibit persistent high levels of anti-PF4-IgG titers for a period of three months or more. Despite receiving standard treatments, these patients may still experience recurrent episodes of thrombocytopenia [30,31]. The persistence of high anti-PF4-IgG titers and the recurrence of thrombocytopenia in such cases suggests a more complex and challenging clinical scenario.

## 4. Conclusions

VITT is a rare but serious complication that has been observed in a small proportion of individuals following COVID-19 vaccination. We have presented a rare case of VITT following BNT162b2 (Pfizer-BioNTech) vaccination, which differs from the previously reported cases associated with adenoviral-vectored DNA vaccines. This case highlights that VITT should still be considered as a possible differential diagnosis for uncommon site thrombosis with thrombocytopenia following mRNA vaccination. Prompt treatment with IVIGs and direct oral anticoagulants can lead to favorable outcomes for patients. Future research should aim to clarify whether the inflammatory stimulus of vaccination induces platelet-activating antibodies and identify which component of the vaccine triggers a new or recalled response to an unrelated host protein, PF4. By investigating these aspects, we can gain a better understanding of vaccine safety and potential rare adverse events. Nevertheless, the public health benefits of universal vaccination far outweigh the occurrence of this rare adverse event.

## Figures and Tables

**Figure 1 vaccines-11-01115-f001:**
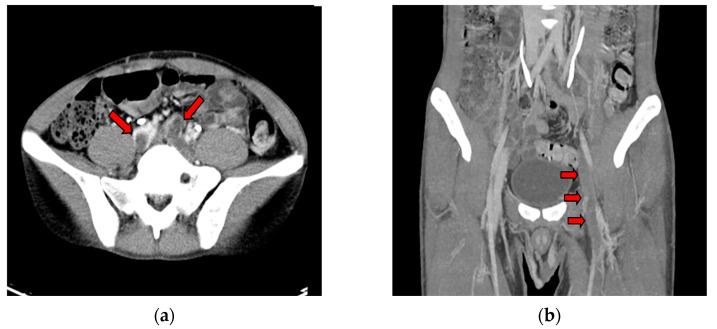
CT venography of bilateral lower extremities showed filling defects (red arrows) within (**a**) bilateral common iliac veins and (**b**) left femoral veins; Brain MRI imaging. (**c**) Axial view—left temporal lobe subacute intracerebral hemorrhage (ICH) with surrounding hyperintensity on T2WI. (**d**) MR venogram showing filling defects in the left transverse sinus and left sigmoid sinus.

**Figure 2 vaccines-11-01115-f002:**
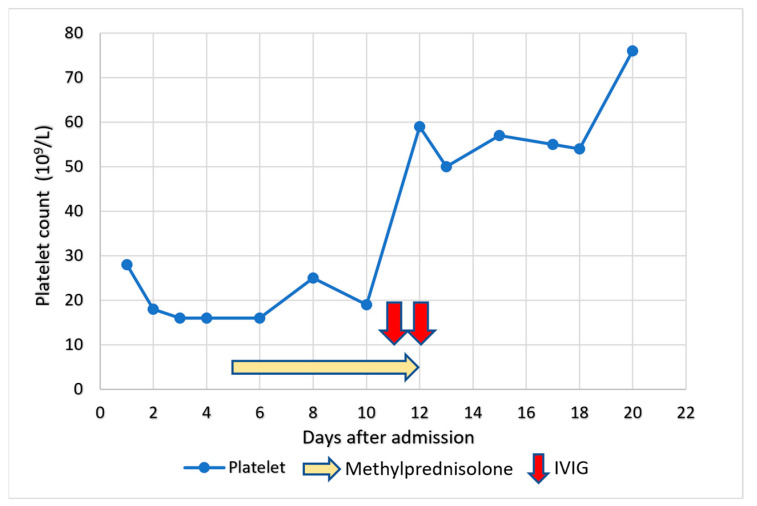
Platelet count during the course of treatment. He had a poor response to steroid use but a prompt response to IVIGs.

**Table 1 vaccines-11-01115-t001:** Laboratory investigations within 30 days after the administration of COVID-19 BNT162b2 vaccine.

Investigation	Result	Unit	Reference Range
Complete Blood Count			
WBC count	10.1	10^9/^L	3.4–9.5
Hemoglobin	15.4	g/dL	13.3–17.2
Platelet count	34	10^9/^L	143–349
MCV	88.6	fL	81.9–98.4
MCH	31.3	pg	27.5–33.7
MCHC	35.3	g/dL	32.8–35.4
RDW	12.5	%	12.0–14.6
APTT	34.2	seconds	29.3–40.1
PT	10.5	seconds	9.4–12.5
WBC Differential Count			
Neutrophils	72.6	%	40.8–76.6
Lymphocytes	15.2	%	15.4–47.0
Monocytes	5.2	%	4.4–11.8
Eosinophils	6.9	%	0.4–7.5
Basophils	0.1	%	0.2–1.7
Biochemical Profile			
Creatinine	0.85	mg/dL	0.70–1.20
AST	37	U/L	10–50
ALT	9	U/L	<50
Fibrinogen	431.3	mg/dL	276–471
FDP(Dimer)	>7650	ng/mL	0–500

## Data Availability

All data available are included in the text. Personal data are not available due to privacy restrictions.

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
