# Peer review of "Vaccine-Induced Immune Thrombotic Thrombocytopenia following BNT162b2 mRNA COVID-19 Booster: A Case Report"

_vaccines, 2023, doi:10.3390/vaccines11061115_

Round 1
Reviewer 1 Report
The authors described a rare case of VITT after administration of the third dose of the Pfizer vaccine. Numerous studies have described the efficacy and safety of anti-COVID-19 vaccines adopted to contrast the recent pandemic. However, large discussion regarding potential adverse events has sometimes negatively affected the screening campaign. The complication known as vaccine-induced immune thrombotic thrombocytopenia (VITT) has been described previously as a rare post-vaccination adverse event in several reports. However, such adverse reactions are usually attributed to adenoviral vector COVID-19 vaccines and, as evidence of this, there are few papers dealing with the development of VITT after the BNT162b2 vaccine. This case-report assumes clinical importance because, in addition to providing information on possible treatment, it underscores the importance of carefully evaluating the choice of vaccine schedule even in patients apparently free of risk factors. The authors have adequately described the methodology for diagnosis of suspected cases of VITT and provided useful suggestions for treatment. Despite the obvious limitations related to case-reports, I confirm the positive opinion regarding the publication aimed to increase the scientific evidence in this field
The topic is relevant, methodology is appropriate, and discussion is aligned with the results.
Minor revision:
1) There is a repetition at line 71 (COVID-19), please remove.
2) The Authors should include one or more references regarding the estimates of VITT development associated to mRNA-vaccine in discussion section, in order to highlight the unquestionable safety and efficacy of COVID19-vaccine.
Author Response
Point 1: There is a repetition at line 71 (COVID-19), please remove.
Response 1: Thank you for this correction. We had removed the repetition in line 75 (page 2).
Point 2: The Authors should include one or more references regarding the estimates of VITT development associated to mRNA-vaccine in discussion section, in order to highlight the unquestionable safety and efficacy of COVID19-vaccine.
Response 2: Thank you for this comment. We revised the sentence in line 152-155 (page 5) to “COVID-19 messenger RNA (mRNA) vaccines have undergone comprehensive assessment in the adult population, revealing their safety and efficacy as potent interventions for mitigating the incidence of SARS-CoV-2 infection, ......”. We also added the related references 21, 22, 23 in line 317-324 (page 8 and 9).
Reviewer 2 Report
The manuscript is a case report of a healthy 28- year-old man who developed Vaccine-induced immune thrombotic thrombocytopenia (VITT) 3 weeks after his third dose of COVID-19 vaccine with first BNT162b2 (from Pfizer-BioNTech) injection. The report describes the serial investigations and confirmed diagnosis by positive PF4 antibody assay (ELISA) followed by treatment with intravenous immunoglobulins (IVIGs).
The manuscript has been written systematically. However, there are some clarifications which need to be addressed:
11. The author confirmed the diagnosis of VITT by measuring the presence of positive anti-PF4 antibodies by enzyme-linked immunosorbent assay. However, elevated anti-PF4 antibodies may not be specific for VITT diagnosis as this assay can be positive in both HIT and VITT. Instead, researchers have developed flow cytometry based platelet activation assay specific to VITT. (Reference: Handtke et al. Blood. 2021 Jul 1; 137(26):3656-3659). The authors are needed to provide justification for using PF4 ELISA.
2. There are other case reports available for mRNA vaccines causing VITT after administration. Like:
a. Chen et al., Front. Neurol. 13:989730. doi: 10.3389/fneur.2022.98973
b. Rodríguez et al., 2021 Dec; 208:1-3. doi:10.1016/j.thromres.2021.10.002
The author should add related references and also discuss relevance of the current study compared to published literature.
33. Are there cases reported in which mRNA vaccines only are administered as prime and booster dose, which had caused VITT? VITT cases are mainly reported in patients primarily vaccinated with ChAdOx1 nCov-19. In current case study, there can be a possibility that prior vaccination with ChAdOx1 nCov-19 vaccine caused persistent low-level/subclinical PF4 antibody response which was aggravated after the booster dose with BNT162b2. Please clarify.
Minor comments:
4. Line 68: write “32 X 103” instead “32 X 103”. Also correct the same in Fig.2 Y axis title.
5. Line71: remove the word “COVID-19”.
Minor edits to be made.
Author Response
Point 1: The author confirmed the diagnosis of VITT by measuring the presence of positive anti-PF4 antibodies by enzyme-linked immunosorbent assay. However, elevated anti-PF4 antibodies may not be specific for VITT diagnosis as this assay can be positive in both HIT and VITT. Instead, researchers have developed flow cytometry based platelet activation assay specific to VITT. (Reference: Handtke et al. Blood. 2021 Jul 1; 137(26):3656-3659). The authors are needed to provide justification for using PF4 ELISA.
Response 1: Thank you for this comment and recommendation. We totally agreed that anti-PF4 antibodies can be positive in both HIT and VITT. In contrast to HIT, platelet activation in VITT occurs in the presence of PF4 rather than low heparin concentrations. Some antibodies that bind to PF4/heparin complexes as detected by EIAs may possess characteristics consistent with typical HIT antibodies. However, our patient has not been exposed to heparin at any point during their lifetime, which eliminates the need for us to differentiate between the two scenarios. Sponetous HIT is a rare condition been reported largely after orthopedic surgery and polyanionic medication. Neither of these conditions observed in our patient. At that time, our country had a policy of sending all samples to a central laboratory for examination, and the testing method employed was Enzyme Immunoassay (EIA). Besides, according to Pavord et al. (N Engl J Med 2021, 385, 1680-1689, doi:10.1056/NEJMoa2109908), the conclusive diagnosis criteria about positive PF-4 antibodies detection was performed using the EIA method.
Point 2: There are other case reports available for mRNA vaccines causing VITT after administration. Like:
- Chen et al., Front. Neurol. 13:989730. doi: 10.3389/fneur.2022.98973
- Rodríguez et al., 2021 Dec; 208:1-3. doi:10.1016/j.thromres.2021.10.002
The author should add related references and also discuss relevance of the current study compared to published literature.
Response 2: Thank you for the comment and recommendation. We added a new paragraph and revised in line 155-168 (page 5) to “Until now, there have been only a limited number of reported instances of VITT attributed to mRNA COVID-19 vaccines, such as mRNA-1273 (Moderna) or BNT162b2. Chen et al. documented a case study detailing VITT characterized by cerebral venous thrombosis and hemorrhage subsequent to the administration of an mRNA-1273 COVID-19 booster. ......” We also added the related reference 24 in line 325-328 (page 9) and reference 25 in line 329-331 (page 9).
Point 3: Are there cases reported in which mRNA vaccines only are administered as prime and booster dose, which had caused VITT? VITT cases are mainly reported in patients primarily vaccinated with ChAdOx1 nCov-19. In current case study, there can be a possibility that prior vaccination with ChAdOx1 nCov-19 vaccine caused persistent low-level/subclinical PF4 antibody response which was aggravated after the booster dose with BNT162b2. Please clarify.
Response 3: Thank you for the comment. We strongly agree with your perspective, and this is an extremely critical issue. That was also our initial question. Rodríguez et al. mentioned a case of VITT administered only mRNA vaccines (BNT162b2) as prime and booster dose. (Thromb Res 2021, 208, 1-3, doi:10.1016/j.thromres.2021.10.002) They believe that in terms of timing, it was caused by the first dose. This case supports that mRNA vaccines themselves can cause VITT. Most platelet-activating anti-PF4 antibodies in VITT are transient, and the median time to negative functional test results was found to be 15.5 weeks. As we mentioned in line 205-208 (page 6), none of 29 VITT patients developed new thrombosis or a rise in anti-PF4/heparin IgG EIA optical density after receiving the mRNA COVID-19 vaccine booster (Blood 2022, 139, 1903-1907, doi:10.1182/blood.2021014214.) Indeed, persistent low-level/subclinical PF4 antibody maybe exist but according to previous study, they seldom cause new thrombosis. In addition, both the timing of the first dose and the second dose of the vaccine (nine and seven months) have exceeded the typical timeframe associated with VITT. Therefore, we still believe that it is most likely the mRNA booster that triggered VITT.
Point 4: Line 68: write “32 X 103” instead “32 X 103”. Also correct the same in Fig.2 Y axis title.
Response 4: Thank you for the comment and correction. We had corrected the word and Y axis title in Fig.2.
Point 5: Line 71: remove the word “COVID-19”.
Response 5: Thank you for this correction. We had removed the word.

Reviewer 3 Report
“Vaccine-induced immune thrombotic thrombocytopenia following BNT162b2 mRNA COVID-19 booster: A case report” is informative manuscript. The manuscript needs revision before publication. I have the following suggestions for authors to address.
1. Check the abbreviations throughout the manuscript and introduce the abbreviation when the full word appears the first time in the text and then use only the abbreviation (For example, VITT, IVIGs, SARS-CoV-2). Make a word abbreviated in the article that is repeated at least two times in the text, not all words to be abbreviated (For example, AVVs).
2. “SARS-CoV-2 virus”---“SARS-CoV-2” (in line 25)
3. “Keywords” should be improved. For example, “SARS-CoV-2” should be removed.
4. In lines 25-27, “The COVID-19 pandemic, resulting from the infection of the SARS-CoV-2 virus, has had a global impact, prompting extensive vaccination as a vital strategy to mitigate its effects.” For the benefits of the readers please list more detailed information (other strategies) for SARS-CoV-2 treatment. For example, “The scientific community has made significant progress in mitigating the threat of COVID-19 through the discovery and development of vaccines (DOI: 10.1016/s0140-6736(22)00055-1), small molecule agents (DOI: 10.1016/j.ejmech.2023.115503), antibodies (DOI: 10.1002/jmv.26789), natural products (DOI: 10.1016/j.jep.2021.113869), and traditional medicines (Cell & Bioscience. 2021, 11, 100).”
5. In Introduction, there are lots of papers about the advantages of vaccines, which should be included in this paper.
6. In “Conclusions” section, the conclusion seems very general, lacking the future aspects. The quality of the conclusion should be improved.
Editing of English language required
Author Response
Point 1: Check the abbreviations throughout the manuscript and introduce the abbreviation when the full word appears the first time in the text and then use only the abbreviation (For example, VITT, IVIGs, SARS-CoV-2). Make a word abbreviated in the article that is repeated at least two times in the text, not all words to be abbreviated (For example, AVVs).
Response 1: Thank you for this comment. We checked the manuscript and abbreviated vaccine-induced immune thrombotic thrombocytopenia as VITT in line 118-119 (page 4), line 170-171 (page 5) and line 191-192 (page 6), 215-216 (page 6), intravenous immunoglobulins as IVIGs in line 115 (page 4), line 194 (page 6) and line 229 (page 6), platelet factor 4 as PF4 in line 135 (page 5) and . We deleted the AAVs abbreviation in line 129 (page 4), line 257-258 (page 7).
Point 2: “SARS-CoV-2 virus”---“SARS-CoV-2” (in line 25).
Response 2: Thank you for this comment. We revised the word in line 25 (page 1).
Point 3: “Keywords” should be improved. For example, “SARS-CoV-2” should be removed.
Response 3: Thank you for this comment and recommendation. We had removed “SARS-CoV-2” and added “COVID-19 vaccine booster” in the keywords in line 22 (page 1).
Point 4: In lines 25-27, “The COVID-19 pandemic, resulting from the infection of the SARS-CoV-2 virus, has had a global impact, prompting extensive vaccination as a vital strategy to mitigate its effects.” For the benefits of the readers please list more detailed information (other strategies) for SARS-CoV-2 treatment. For example, “The scientific community has made significant progress in mitigating the threat of COVID-19 through the discovery and development of vaccines (DOI: 10.1016/s0140-6736(22)00055-1), small molecule agents (DOI: 10.1016/j.ejmech.2023.115503), antibodies (DOI: 10.1002/jmv.26789), natural products (DOI: 10.1016/j.jep.2021.113869), and traditional medicines (Cell & Bioscience. 2021, 11, 100).”
Response 4: Thank you for this comment and recommendations. We very much appreciate this suggestion and we added the sentence in line 27-30 (page 1) “In order to control the pandemic, the scientific community has made significant pro-gress in mitigating the threat of COVID-19 through the discovery and development of vaccines [1], small molecule agents [2], antibodies [3], natural products [4], and tradi-tional medicines [5].” We also added the related references [1]-[5] in line 262-274 (page 7).
Point 5: In Introduction, there are lots of papers about the advantages of vaccines, which should be included in this paper.
Response 5: Thank you for this recommendation. We added several papaers about the safety and efficacy regarding COVID-19 vaccines. We revised the reference in line 31 (page 1) to “[6-10] ” and added the new references 3-5 in line 267-274 (page 7).
(Baden et al. Efficacy and Safety of the mRNA-1273 SARS-CoV-2 Vaccine, doi:10.1056/NEJMoa2035389; Voysey et al. Single-dose administration and the influence of the timing of the booster dose on immunogenicity and efficacy of ChAdOx1 nCoV-19 (AZD1222) vaccine: a pooled analysis of four randomised trials, doi:10.1016/S0140-6736(21)00432-3; Sadoff, J et al. Safety and Efficacy of Single-Dose Ad26.COV2.S Vaccine against Covid-19, doi:10.1056/NEJMoa2101544)
Point 6: In “Conclusions” section, the conclusion seems very general, lacking the future aspects. The quality of the conclusion should be improved.
Response 6: Thank you for this comment and recommendation. We think the future aspects may focus on whether these antibodies are autoantibodies generated by the inflammatory stimulus of vaccination or by the vaccine itself. We revised the sentence in line 229-233 (page 6) to “Future research should aim to clarify whether the inflammatory stimulus of vaccination induces platelet-activating antibodies and identify which component of the vaccine trigger a new or recalled response to an unrelated host protein, PF4. By investigating these aspects, we can gain a better understanding of vaccine safety and potential rare adverse events.”
Round 2
Reviewer 2 Report
The manuscript reads well and can be accepted for publication.